

# Measuring the attenuation length of muon number in the air shower with muon detectors of 3/4 LHAASO array

**Xiaoting Feng[1],⋆, Hengying Zhang[2],†, Cunfeng Feng[1] and Lingling Ma[2]**
**on behalf of the LHAASO Collaboration**

**1** Institute of Frontier and Interdisciplinary Science, Shandong University, Qingdao, China
**2** Institute of High Energy Physics, Chinese Academy of Sciences, Beijing, China

⋆ fengxt@mail.sdu.edu.cn, † hyzhang@ihep.ac.cn

## Abstract

LHAASO KM2A consists of 5915 scintillation detectors and 1188 muon detectors, and the muon detectors cover 4% area of the whole array with 30 m interdetector spacing. The muon number in air shower events, with very high energy, is investigated with the data recorded by muon detector of the 3/4 LHAASO array in 2021. The attenuation length of muon number in the air shower is measured by fitting the muon number with constant flux in various zenith angles, based on the constant intensity cut method. The variation of the attenuation length as shower energy from hundreds TeV to tens PeV is presented. The results of simulation also is presented for comparing.

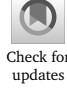

## 1 Introduction

Muons are basically produced in the decay of shower hadrons, such as charged pions and kaons, during the early processes of the EAS (extensive air shower). They are very penetrating particles and suffer less attenuation in the atmosphere than electromagnetic and hadronic components of showers. Thus, air shower muons keep the original information of hadronic interaction at very high energies and hence the mean number of muons in a shower could be used to validate hadronic interaction models by comparing EAS predictions with experiment data. On the other hand, the muon number is also sensitive to the mass composition of the cosmic ray, since the mean number of muons produced at one and the same interaction energy grows in accordance with the atomic mass *A* of interacting particles [1].

The HiRes/MIA collaboration already reported a discrepancy in simulated and measured muon number in air showers between $10^{17}$ to $10^{18}$ eV in the year 2000 [2]. A combined

analysis of eight experiments (EAS-MSU, IceCube, KASCADE-Grande, NEVOD-DECOR, Pierre Auger, SUGAR, Telescope Array, and Yakutsk) shows that the muon deficit in simulation increases with the energy [3]. Recently experiment results also show that the muon deficit is greater at larger distances to the shower axis [4], and the attenuation of the muon content measured is lower than that of the predicted [5].

The LHAASO (Large High Altitude Air Shower Observatory) experiment has 1178 buried water Cherenkov detectors over one square kilometer area with 30 m interdetector spacing. It provides one new facility to measure the muon content in air shower around knee region of cosmic rays. In this paper, we will present the preliminary results on the attenuation length of muon number in the EAS measured with the muon detectors in the 3/4 LHAASO KM2A.

## 2 LHAASO muon detector and muon number measurement

LHAASO is a ground-based air shower observatory located at 4410 m above mean sea level in Daocheng, China [6]. It is a hybrid detectors array consists of an EAS array covering an area of 1.3 km$^2$ (KM2A), 78,000 m$^2$ water Cherenkov detector array (WCDA) and 18 wide-field air Cherenkov/fluorescence telescopes (WFCTA). The KM2A consists of muon detectors (MDs) with a interdetector spacing of 30 m and electromagnetic detectors (EDs) with a interdetector spacing of 15 m which record the muon and electromagnetic particles respectively.

In this paper, the analysis is based on the data sample collected from January to June of 2021 with the 3/4 LHAASO KM2A, which includes 3978 EDs and 917 MDs as shown in the left plot of Figure 1.

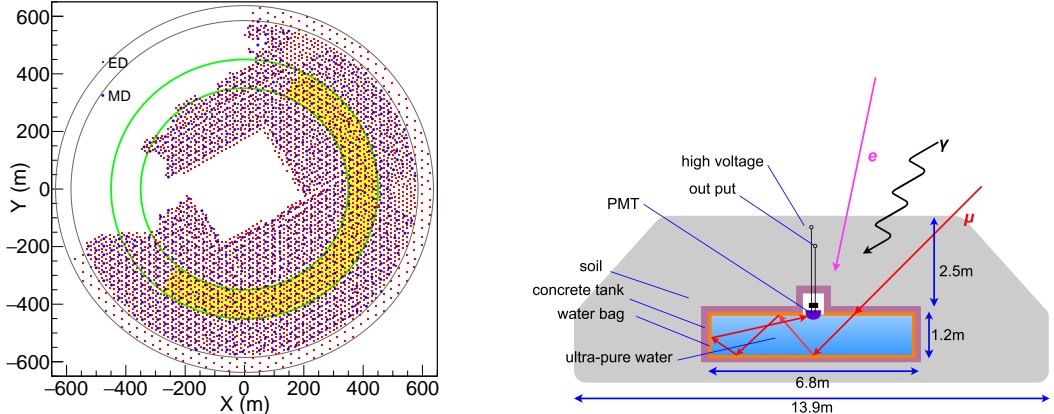

Figure 1: Left: Layout of 3/4 LHAASO KM2A. The yellow area show the core distribution of events after selection. Right: Schematic of LHAASO muon detector.

### 2.1 Muon detector and muon measurement

The effective area of one MD is 36 m$^2$. The total muon sensitive area in LHAASO is more than 40,000 m$^2$. The MD has a wide dynamic range up to 10,000 particles, which enables to measure the muon content in a large energy range without saturation [6]. The right plot of Figure 1 shows the schematic of MD. The housing of unit MD is a concrete tank whose diameter is 6.8 m and depth is 1.2 m. Tank internal is a water bag with reflectivity higher than 95%, and ultra-pure water is enclosed by tank. The thickness of the overburden soil above MD is 2.5 m that absorb electromagnetic and other charged particles in the shower, the muons with the energy above 1 GeV can pass through the overburden soil. At the top of the water is

an 8-inch photo-multiplier tube (PMT), PMT collects the Cherenkov light which yield by high speed muons.

The PMT signals are digitized by flash analog-to-digital converters (FADCs) [6]. When signals of one FADC channel amplitude exceeds the preset threshold, the signal is recorded as one 'hit' and this MD is recorded as fired detector. A time stamp of the hit is generated by a time-to-digital converter (TDC). Hits from MDs and EDs in a time window of $\pm 5$ $\mu s$ around the event trigger time are collected and built into one event. Only the ED and MD hits within [-30,50] ns of the shower front plane are selected for further analysis work. An event trigger is generated if the hit multiplicity in the time window of 200 ns exceeds a preset multiplicity threshold.

The number of muon for one triggered event is equal to the integral charge of all the hits divided by the VEM (vertical equivalent muon). In order to reduce the punch-through effect of the high energy particles near the shower core, all the hits within 40 m from the shower axis will not be counted. Also, the hits far from 200 m of the shower core are not counted also due to the limited geometry of the KM2A. So, in this paper, the number of muon $N_\mu$ in one shower event is defined as the total muons that are obtained by counting the MD within the $40-200$ m ring from the shower axis. For the inclined shower with zenith angle $\theta$, the number of muon $N_\mu$ will be corrected.

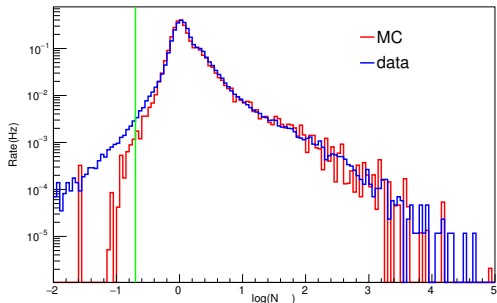
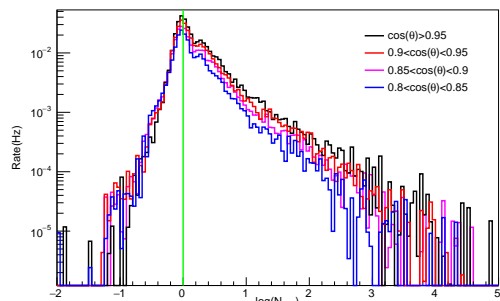

Figure 2: The hit rate distribution of one single MD with respect to the muon number of the hit $N_{\mu\_u}$. Left plot shows the simulation result together with the data, which match well above 0.2 muons as indicated by the green vertical line. Right plot shows the hit rate for various zenith angles, where the single muon peak is same for all the zenith angles as indicated by the vertical green line.

## 2.2 Simulation

The COsmic Ray SImulations for KAscade (CORSIKA) code (version7.6400) [7] software package is used to simulate EASs of cosmic ray with primary energy from 100 TeV to 10 PeV. The zenith angle of incident cosmic ray were sampled within $0°-70°$. In CORSIKA simulation, both hadronic interaction models QGSJET-II-04 and EPOS-LHC are applied for model checking. Primary particle components contained in the simulation include hydrogen (H), helium (He), nitrogen (N), aluminum (Al) and iron (Fe) nuclei. The detector response of ED and MD were simulated by G4KM2A [8] package which was developed in the framework of GEANT4. The random noise in the single of ED and MD is also considered in the simulation.

After the simulation data is normalized according to the H3a model spectrum of [9]. The rate of hits for one single MD is shown in the left plot of Figure 2 together with the experiment data. The simulated hit rate is matched well with the data except below the 0.2 muons. So only the hit larger than 0.2 muons will be counted for the number of muon $N_\mu$ in the shower event. The hit rate distribution for various zenith angle intervals is shown in the right plot of

Figure 2. It's clear to find that the single muon peak is equal for various zenith angle after this correction.

## 2.3 Event selection

The following event selection criteria are applied in this work: (1) NfiltE $\geq$ 50 (NfiltE: the hit number of EDs after filter out noise); (2) NfiltM $\geq$ 15 (NfiltM: the hit number of MDs after filter out noise); (3) NtrigE $\geq$ 50 (NtrigE: the EDs number after filter out noise); (4) 350 m $\leq R_p \leq$ 450 m ($R_p$: distance from shower axis to array center); (5) zenith angel $\theta \leq 45°$.

After application the above event selection criteria, more than $5 \times 10^8$ events left and the shower core of these events distribution is shown in left plot of Figure 1. More than $3.7 \times 10^5$ simulation events pass the selection criteria. The event rate distribution for the data and simulation is shown in Figure 3, and event ratio, the histograms of both models divided by data, are shown in the bottom plot of Figure 3, and the ratio plot indicates the data matched well with the simulation.

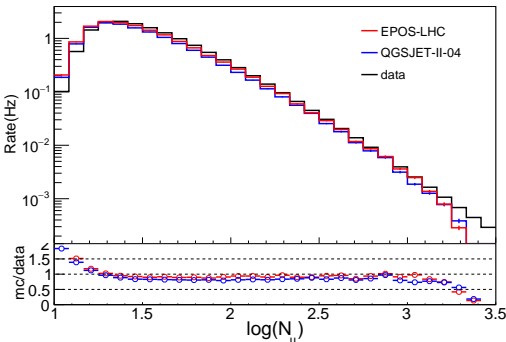

Figure 3: Top: The event rate distribution with the number of muon $N_\mu$ after event selection for simulation samples and data. Bottom: The event rate ratio of simulation (EPOS-LHC: red; QGSJET-II-04: blue) over data bin by bin.

## 3 Attenuation length of muon number

According the isotropic assumption of cosmic ray, the same intensity corresponds to the same primary energy for showers of cosmic rays at various zenith angle. The muon number at the same intensity varies with zenith angle $\theta$, because the showers travel through different path for different zenith angle and the evolution of muon number in the atmosphere is different. For the showers with same primary energy, formula relationship of muon number and zenith angle follows formula (1):

$$N_\mu(\theta) = N_\mu^0 e^{\frac{-X_0 \sec\theta}{\Lambda_\mu}}, \tag{1}$$

where $N_\mu(\theta)$ is the muon number in the shower with the zenith angle $\theta$, $N_\mu^0$ is the normalization parameter. $X_0$ is the vertical atmospheric depth, which is $600 g/cm^2$ at the LHHASO level, and the $\Lambda_\mu$ is the attenuation length of muon number.

After event selection according above criteria, the events within the zenith angle from $0° - 45°$ were grouped into 5 zenith angle intervals with the same aperture. The integral muon spectra for each zenith angle are shown in the left of Figure 4. Fifteen cuts are applied on $J(> N_\mu, \theta)$ at different constant integral intensities in order to select showers with the

same frequency rate at distinct zenith angles. This procedure is performed within the interval $logJ \in [-6, -4.3]$ of full efficiency and maximum statistics as shown in the left plot of Figure 4. Each cut will cross with these five integral muon spectra, the corresponding $N_\mu$ for these five intersection points is plotted in the right plot of Figure 4. Each group is fitted with formula (1) as the line in the plot. The slope of the fitted line is equal to $\frac{X_0}{\Lambda_\mu}$.

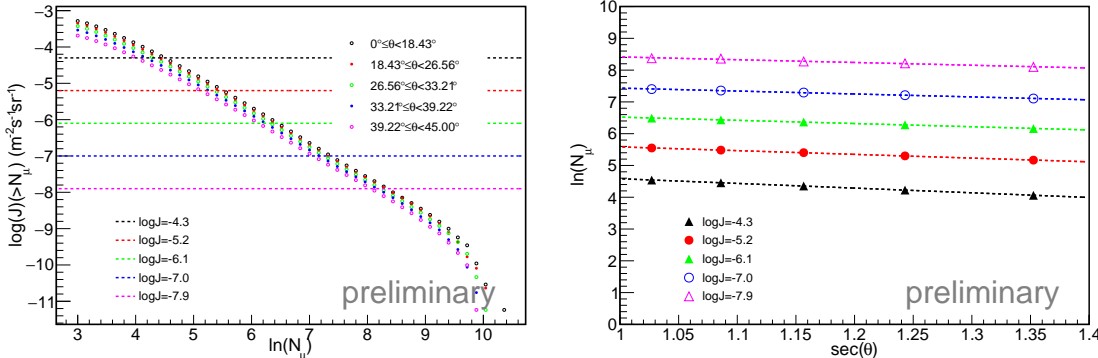

Figure 4: Left: The integral flux spectra of the muon number $N_\mu$ of the shower, only statistic error shown. Only 5 of 15 cuts are shown. Right: The muon number $N_\mu$ of the shower various with the zenith angle $\theta$.

According to the fitting result of right plot of Figure 4 at each flux intensity, the attenuation length of muon number are shown in left plot of Figure 5. The attenuation length is found to decrease as the intensity increases. The intensity corresponds to shower energy, which can be reconstructed from $N_\mu$ of vertical shower. So, the attenuation length increases with the shower energy as shown in the right plot of Figure 5.

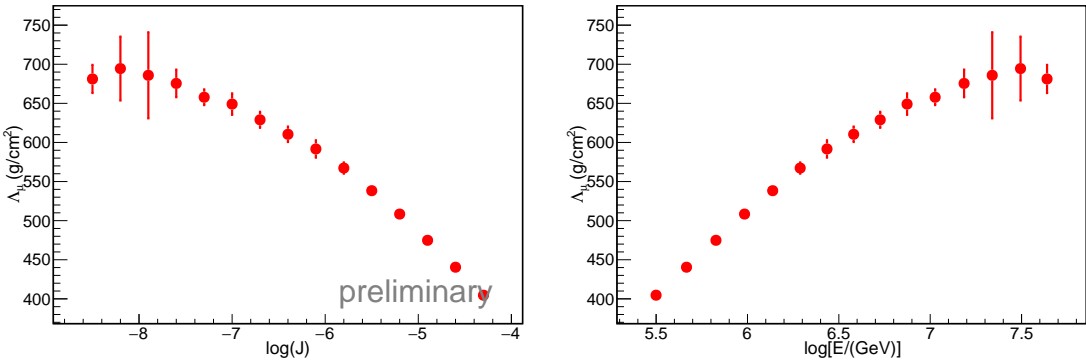

Figure 5: Left: The attenuation lengths of muon number various with the flux $J(> N_\mu, \theta)$, only statistics error shown. Right: Attenuation lengths various with shower energy.

The attenuation length of muon number is also measured with the method for simulation data. As shown in Figure 6, the simulation result for both hadronic models QGSJET-II-04 and EPOS-LHC are consistent well with the experiment predicted with the energy below 1 PeV. The attenuation length of simulation also increases with shower energy. The statistic of the simulation for shower with energy above PeV is very limited, it should be improved in the future for the full LHAASO array analysis work.

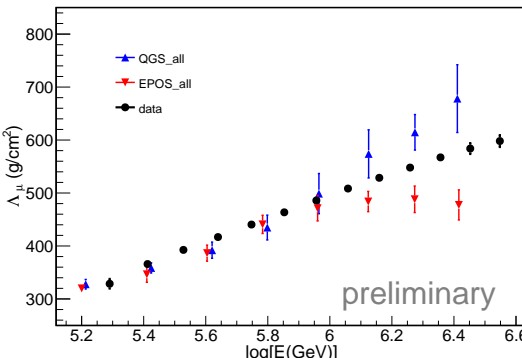

Figure 6: The $\Lambda_\mu$ of Data and MC. Error bar indicate statistical error.

## 4 Conclusion

This work is based on the data collected during January to June of 2021 with the 3/4 LHAASO-KM2A operation. By comparing with the simulation results, the muon number measured by LHAASO muon detector is agreed well with expected results for various zenith angle shower. The event rate distribution is matched with the expected results also.

The attenuation length of muon attenuation in the air shower is measured with 3/4 LHAASO-KM2A based on the constant intensity cut method. The preliminary results of attenuation length are presented for shower energy above hundreds TeV to tens of PeV, where the increasing trend of the attenuation length with the shower energy is clear. This trend also is found in the simulation results for both hadronic interaction model EPOS-LHC and QGSJET-II-04.

## Acknowledgements

**Funding information**    This work is supported in China by NFSC (No. 12175121, U1931108), Shandong Provincial Natural Science Foundation(No.ZR2019MA014 ) and Young Scholars Program of Shandong University (No. 2018WLJH78).

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
