# Peer review of "Measuring the attenuation length of muon number in the air shower with muon detectors of 3/4 LHAASO array"

_SciPost Physics Proceedings, doi:SciPost Phys. Proc. 13, 017 (2023)_

## Round 1 · Referee Report · Anonymous (Referee 1) · 2022-9-19

# Reviewer's Report

Date: 19/09/2022

Manuscript Title : **Measuring the attenuation length of muon number in the air shower with muon detectors of 3/4 LHAASO array**

Authors : **Xiaoting Feng, Hengying Zhang, Cunfeng Feng, Lingling Mab (on behalf of the LHAASO Collaboration)**

In this paper authors reported the preliminary results of the study of the attenuation length of muons in EAS using the data of muon detectors of 3/4 LHAASO array. The results look interesting, to be useful in the various aspects of studies of cosmic rays in future and hence the paper can be published in the proceedings of ISVHECRI 2022. However, before publish the paper it needs extensive language editing and few technical corrections. Some of them which I have notice are as follows:

**Title:**

I think it would be more appropriate if authors replace the term "**attenuation length of muon number**" by the "**attenuation length of muons**". This change should be applied in the whole body of the paper.

**Abstract:**

1) In line 2, I would suggest replacing the term "**with 30 m spacing**" by "**with 30 m inter-detector spacing**". This is applicable to the whole paper.

2) In line 2, it would be more appropriate if the word "**of**" after "**muon number**" is replaced by the word "**in**".

**Introduction:**

1) In the first line authors wrote that muons are produced in the decay of baryons and mesons. But what I know is that in cosmic rays muons are basically produced from the decays of mesons, mostly in the decays of pions and kaons. Authors should clarify this issue, otherwise the sentence should be corrected.

2) In line 4, the word "**air**" should be written before the world "**shower**". The term "**and hence**" should be written after the word "**energies**" after replacing the existing comma.

3) In page 1, last paragraph, line 4, the word "**increasing**" should be replaced by the word "**increases**".

4) The last sentence in the page 1 should be modified as "**Recent experiment results also show that the muon deficit is greater at larger distances to the shower axis [4], and the attenuation of the muon content measured is lower than that of the predicted [5].**"

5) In page 2, first paragraph, line 1, the word "**The**" is required before word "**LHAASO**" and the word "**built**" should be replaced by the word "**has**".

6) In page 2, first paragraph, line 3, the term "**cosmic ray** should be replaced by "**cosmic rays**". This should be followed in the rest of the paper appropriately.

**Section 2:**

1) In paragraph 1, line 1, the term "**above sea level**" should be replaced by the proper one "**above mean sea level**". In line 2, the word "**that**" in front of "**consists**" should be removed. In lines 5 and 6, the term "**which will record**" should be replaced by the "**which record**".

2) In paragraph 2, line 1, the month "**Jan**" should be written in the full form as "**January**".

**Subsection 2.1:**

1) In paragraph 1, line 4, the term "**which diameter is 6.8 m**" should be corrected as "**whose diameter is 6.8 m**".

2) In page 3, paragraph 2, line 4, the sentence "**On the other hand, the hits 200 m far from the shower core isn't counted also....**" should be rewritten as "**Also, the hits far from 200 m of the shower core are not counted...**". In line 6, the term "**the total muons by counting**" should be modified by "**the total muons that are obtained by counting**".

**Subsection 2.2:**

1) In paragraph 1, line 2, the term "**EAS of cosmic ray**" should be modified as "**EASs of cosmic rays**". In line 5, the word "**the**" would be appropriate before the word "**simulation**". In line 7, the term "**which developed**" should be corrected as "**which was developed**" and the term "**noise in single**" should be corrected as "**noise in the single of**".

2) In paragraph 2, line 1, the term "**data normalized**" should be corrected as "**data is normalized**". The comma after the Ref. "**[9]**" should be replaced by full stop and the next line should be changed accordingly. In line 3, the sentence "**The hit rate is matched well between simulation and data except below the 0.2 muons.**" should be modified as "**The simulated hit rate is matched well with the data except below the 0.2 muons.**". In line 5, the word "**angle**" is required after the word "**zenith**" with the word "**interval**" as "**intervals**". In the last line no comma is required after the word "**that**".

**Figure 2:**

1) What is $N_{\mu\_u}$? I think it should be $N_\mu$. So the title of the $x$-axis of both the plots should be corrected.

2) The caption of the figure should be corrected as "**The hit rate distribution of one single MD with respect to the muon number of the hit $N_\mu$. Left plot shows the simulation result together with the data, which match well above 0.2 muons as indicated by the green vertical line. Right plot shows the hit rate for various zenith angles, where the single muon peak is same for all the zenith angles as indicated by the vertical green line.**".

3) Most importantly, it is clear from the figure that the vertical green line of the left plot is misplaced. It should pass through $\log(N_\mu) \sim -0.4$, which corresponds to $N_\mu \sim 0.4$. That is the simulated hit rate and data match well above the $0.4$ muons only not $0.2$ muons. Therefore all related analysis and texts including the caption of the figure are to be corrected accordingly.

4) For clarity the legend of the right plot should be enlarged.

**Subsection 2.3:**

1) In paragraph 2, line 2, the term "**of these events**" should be added after the word "**distribution**". In line 4-5, the term "**selection, the event rate of the data is also plotted together with simulation**" should be written as "**selection together with the event rate data**". The last sentence is not clear and should be written clearly. what is ratio plot? How such ratios are obtained.

**Figure 3:**

1) The bottom plot of this figure is not clear. It should be drawn as a separate plot.

2) The second sentance in the caption of the figure is not clear. Should be written clearly.

**Section 3:**

1) In paragraph 1, line 1, there is an extra "**the**" after the first word. In line 3, the term "**of cosmic rays**" should be added before the word "**varies**". In line 4, the word "**shower**" should be replaced by "**showers**". In line 5, the word "**the**" is required before the word "**formula**".

2) A full stop is required at the end of equation (1).

3) For the better readability the sentences below equation (1) should modified as "**Where $N_\mu(\theta)$ is the muon number in the shower with the zenith angle $\theta$, $N_\mu^0$ is the normalization parameter, $X_0$ is the vertical atmospheric depth, which is $600\,g/cm^2$ at the LHHASO level, and the $\Lambda_\mu$ is the attenuation length of muons.**"

4) It is not clear how tha constant intensity lines are plotted in Fig. (4). Process of obtaining these lines should be written in the text of the paper. What is "**lg J**" in this figure ? I think it is "**log J**" and should be like this.

5) In page 4, last paragraph, the first line is not clear. Should be written clearly.

6) The caption of the Fig. 5 should be written correctly and clearly.

7) Below Fig. 5, in line 2 what is "**cutting integral intensity** ? In lines 2-3, the sentence "**The attenuation length decrease as the intensity increase.**" should be written as "**The attenuation length is found to decrease as the intensity increases.**". Similarly, the last sentence of the paragraph beginning from line 5 should be checked and to be made readable.

8) In page 5, last paragraph, in line 1, the word "**the**" would be appropriate before the word "**same**". Similarly in line 3, the word "**the**" would be appropriate before the word

"**LHAASO**". In the line before the last line, the word "**for**" would be appropriate before the word "**full**".

9) In the caption of Fig. 6, the word "**indict**" should be replaced by the word "**indicate**" or "**represent**" appropriately in its locations.

**Section 4:**

1) In paragraph 2, line 2, what is "**CIC** ? I think it means "**Constant Intensity Cut**", but this abbreviation is not mentioned earlier. In line 3, the term "**TeV to tens PeV, the increasing trend as the shower energy is clear**" should be rewritten as "**TeV to tens of PeV, where the increasing trend of the attenuation length with the shower energy is clear**".

** Overall, the clarity of all figures should also be increased.

---

## Round 2 · Referee Report · Anonymous (Referee 2) · 2022-9-28

Report

It is observed that in this version authors have only slightly implemented my previous report on the paper. Still I stand with my whole previous report. Authors of this paper must respect and implement fully and
effectively my previous report for recommending the paper to be published in
the proceedings of ISVHECRI 2022.

Attachment

  • validity: -
  • significance: -
  • originality: -
  • clarity: -
  • formatting: -
  • grammar: -

Author:  Xiaoting Feng  on 2022-10-06  [id 2881]

(in reply to Report 1 on 2022-09-28)

Thank you for your report, we thank the reviewers for the time and effort that they have put into reviewing the previous version of the manuscript. Their suggestions have enabled us to improve our work. Based on the instruction provided in your review report, we uploaded the file of the revised manuscript, and we are point-to-point response to the comments. The comments are reproduced and our responses are given directly afterward in a different color(red).

Attachment:

reply_to_report.pdf

---

## Round 3 · Referee Report · Anonymous (Referee 3) · 2022-10-9

Report

Authors have implemented most of my comments/suggestions in this version of the paper. Also I'm almost satisfied with their responses. So, this version of the paper can be accepted for publication.

---

## Round 3 · Author Response

Thank you for your report, we thank the reviewers for the time and effort that they have put into reviewing the previous version of the manuscript. Their suggestions have enabled us to improve our work. Based on the instruction provided in your review report, we uploaded the file of the revised manuscript, and we are point-to-point response to the comments. The comments are reproduced and our responses are given directly afterward.

---

## Round 3 · List of Changes

Title:
I think it would be more appropriate if authors replace the term “attenuation length of muon number” by the “attenuation length of muons”. This change should be applied in the whole body of the paper.
Response: Thanks for pointing it out. We learn this concept from the KASCADE-Grand paper Astroparticle Physics, vias this link https://doi.org/10.1016/j.astropartphys.2017.07.001. We measured the changing of muon number in shower instead of one muon, or in other words, we measured the collective of shower muon instead of one muon. So, we like to keep the muon number in this manuscript. Thanks for your understanding and supporting to us.
Abstract:
In line 2, I would suggest replacing the term “with 30 m spacing” by ”with 30 m interdetector spacing”. This is applicable to the whole paper.
In line 2, it would be more appropriate if the word “of” after ”muon number” is replaced by the word “in”.
Response: Thank you so much for your careful check, and the mistake has been corrected in the revised manuscript.
Introduction:
In the first line authors wrote that muons are produced in the decay of baryons and mesons. But what I know is that in cosmic rays muons are basically produced from the decays of mesons, mostly in the decays of pions and kaons. Authors should clarify this issue, otherwise the sentence should be corrected.
Response: We are grateful for the suggestion. The muon is mainly produced by the meson decay, but the baryon decay can also produce the muon. To be more clearly and in accordance with the reviewer concerns, the sentence “muons are produced in the decay of baryons and mesons” in the first line modified as “Muons are basically produced in the decay of shower hadrons, such as charged pions and kaons, during the early processes of the EAS (extensive air shower).”
In line 4, the word “air” should be written before the world “shower”. The term “and hence” should be written after the word “energies” after replacing the existing comma.
Response: Thank you for the suggested. The precedent version of the term has been replaced.
In page 1, last paragraph, line 4, the word “increasing” should be replaced by the word “increases”.
Response: Thank you so much for your careful check, and the mistake has been corrected.
The last sentence in the page 1 should be modified as “Recent experiment results also show that the muon deficit is greater at larger distances to the shower axis [4], and the attenuation of the muon content measured is lower than that of the predicted ⑸・”
Response: Thank you for the suggested. The precedent version of the sentence has been modified.
In page 2, first paragraph, line 1, the word “The” is required before word “LHAASO” and the word “built” should be replaced by the word “has”.
Response: Thank you so much for your careful check, and the mistakes have been corrected.
In page 2, first paragraph, line 3, the term “cosmic ray should be replaced by “cosmic rays”. This should be followed in the rest of the paper appropriately.
Response: Thank you so much for your careful check, and the mistake has been corrected.
Section 2:
In paragraph 1, line 1, the term “above sea level” should be replaced by the proper one “above mean sea level”. In line 2, the word “that” in front of “consists” should be removed. In lines 5 and 6, the term “which will record” should be replaced by the “which record”.
In paragraph 2, line 1, the month “Jan” should be written in the full form as “January”.
Response: Thank you so much for your careful check, and the mistakes in paragraph 1 and paragraph 2 have been corrected.
Subsection 2.1:
In paragraph 1, line 4, the term “which diameter is 6.8 m” should be corrected as “whose diameter is 6.8 m”.
Response: Thank you for the suggestion, we have corrected this term.
In page 3, paragraph 2, line 4, the sentence “On the other hand, the hits 200 m far from the shower core isn't counted also ” should be rewritten as “Also, the hits far from 200 m of the shower core are not counted…”.In line 6, the term “the total muons by counting” should be modified by “the total muons that are obtained by counting”.
Response: Thank you for the suggestion, we have rewritten the sentences.
Subsection 2.2:
In paragraph 1, line 2, the term “EAS of cosmic ray” should be modified as “EASs of cosmic rays”. In line 5, the word “the” would be appropriate before the word “simulation”. In line 7, the term “which developed” should be corrected as “which was developed” and the term “noise in single” should be corrected as “noise in the single of”.
Response: Thank you so much for your careful check, and the mistakes in paragraph 1 have been corrected.
In paragraph 2, line 1, the term “data normalized” should be corrected as “data is normalized”. The comma after the Ref. “[9]” should be replaced by full stop and the next line should be changed accordingly. In line 3, the sentence “The hit rate is matched well between simulation and data except below the 0.2 muons.” should be modified as “The simulated hit rate is matched well with the data except below the 0.2 muons.”. In line 5, the word “angle” is required after the word “zenith” with the word “interval” as “intervals”. In the last line no comma is required after the word “that”.
Response: The mistakes in paragraph 2 have been corrected.
Figure 2:
What is N_(μ_u)? I think it should be N_μ. So the title of the x-axis of both the plots should be corrected.
Response: We are so grated for your kind question, I am sorry that this part was not clear in the original manuscript. N_μ in the article represented the total muon number in the shower, and N_(μ_u) in this article represented the total muon number by the unit muon detector (MD), N_μ is the sum of N_(μ_u) detected by multiple MDs. In Figure 2 we want to expression that muon number which detected by single MD are match well. Description on it has been added in the manusciption.
The caption of the figure should be corrected as “The hit rate distribution of one single MD with respect to the muon number of the hit N_μ. Left plot shows the simulation result together with the data, which match well above 0.2 muons as indicated by the green vertical line. Right plot shows the hit rate for various zenith angles, where the single muon peak is same for all the zenith angles as indicated by the vertical green line.”.
Response: Thank you for the suggestion, we have corrected the caption, and the caption of the figure be corrected as “The hit rate distribution of one sin¬gle MD with respect to the muon number of the hit N_(μ_u). Left plot shows the simulation result together with the data, which match well above 0.2 muons as indicated by the green vertical line. Right plot shows the hit rate for various zenith angles, where the single muon peak is same for all the zenith angles as indicated by the vertical green line.”
Most importantly, it is clear from the figure that the vertical green line of the left plot is misplaced. It should pass through log⁡(N_μ)~-0.4, which corresponds to N_μ~-0.4. That is the simulated hit rate and data match well above the 0.4 muons only not 0.2 muons. Therefore all related analysis and texts including the caption of the figure are to be cor¬rected accordingly.
Response: We are so grated for your kind question, however in the left plot of Figure 2, the vertical green line is placed pass through log⁡(N_(μ_u))~-0.7, which corresponds to N_(μ_u)~0.2, so the caption of the figure be corrected as “which match well above 0.2 muons as indicated by the green vertical line.” not 0.4 muons.
For clarity the legend of the right plot should be enlarged.
Response: The legend of the right plot has been enlarged.
Subsection 2.3:
1) In paragraph 2, line 2, the term “of these events” should be added after the word “distribution”. In line 4-5, the term “selection, the event rate of the data is also plotted together with simulation” should be written as “selection together with the event rate data”. The last sentence is not clear and should be written clearly. what is ratio plot? How such ratios are obtained.
Response: The last sentence has been written as “The event rate distribution for the data and simulation sample is shown in Figure 3, and event rate ratio, the histograms of both models divided by data, are shown in the bottom plot of Figure 3”. The ratio plot indicates the event rate histogram of Monte Carlo simulation (MC) divides data histogram.
Figure 3:
The bottom plot of this figure is not clear. It should be drawn as a separate plot.
Response: In the top plot of Figure 3, these are the event rate histograms of data and simulation, and in the bottom plot, the points are the ratio of these event rate, in order to keep x-axis consistent for comparison, so we draw the event histograms and ratio in the same plot. we have enlarged the Figure to ensure the it can be seen clearly.
The second sentence in the caption of the figure is not clear. Should be written clearly.
Response: The second sentence in the caption has been rewritten as “The event rate ratio of simulation (EPOS-LHC: red; QGSJET-II-04: blue) over data bin by bin”.
Section 3:
In paragraph 1, line 1, there is an extra “the” after the first word. In line 3, the term “of cosmic rays” should be added before the word “varies”. In line 4, the word “shower” should be replaced by “showers”. In line 5, the word “the” is required before the word “formula”.
Response: The mistakes in the paragraph have been corrected.
A full stop is required at the end of equation (1).
Response: The full stop has been added at the end of equation.
For the better readability the sentences below equation (1) should modified as “Where N_μ (θ)is the muon number in the shower with the zenith angle θ, N_μ^0 is the normal¬ization parameter, X_0。is the vertical atmospheric depth, which is 600g/〖cm〗^2, at the LHHASO level, and the Λ_μis the attenuation length of muons.”
Response: Thank you for the suggestion, we have rewritten the sentences.
It is not clear how the constant intensity lines are plotted in Fig. (4). Process of obtain¬ing these lines should be written in the text of the paper. What is “lg J” in this figure 4? I think it is “log J” and should be like this.
Response: In order to clear how the constant intensity lines are plotted, we added content in this article as “Fifteen cuts are applied on J(>N_μ,θ) at different constant integral intensities in order to select showers with the same frequency rate at distinct zenith angles. This procedure is performed within the interval logJ∈[-6,-4.3] of full efficiency and maximum statistics as shown in the left plot of Figure 4. Each cut will cross with these five integral muon spectra, the corresponding N_μ for these five intersection points is plotted in the right plot of Figure 4.”
In page 4, last paragraph, the first line is not clear. Should be written clearly.
Response: In order to make readers understand better, we combine this paragraph with the previous one.
The caption of the Fig. 5 should be written correctly and clearly.
Response: The caption of Figure 5 has been written as “Left: The attenuation lengths of muon number various with the flux J(>N_μ,θ), only statistics error shown. Right: The attenuation lengths various with shower energy.”
Below Fig. 5, in line 2 what is “cutting integral intensity”? In lines 2-3, the sentence “The attenuation length decreases as the intensity increase.” should be written as “The attenuation length is found to decrease as the intensity increases.”. Similarly, the last sentence of the paragraph beginning from line 5 should be checked and to be made read¬able.
Response: “cutting integral intensity” is the y-axis of Figure 5, in order to understand better, we have corrected the first sentence as “According to the fitting result of right plot of Figure 4 at each flux intensity, the attenuation length of muon number are shown in left plot of Figure 5. The attenuation length is found to decrease as the intensity increases”, and the last sentence of this paragraph has been rewritten as “So, the attenuation length increases with the shower energy as shown in the right plot of Figure 5”
In page 5, last paragraph, in line 1, the word “the” would be appropriate before the word “same”. Similarly, in line 3, the word “the” would be appropriate before the word “LHAASO”. In the line before the last line, the word “for” would be appropriate before the word “full”.
Response: The mistakes in the paragraph have been corrected.
In the caption of Fig. 6, the word “indict” should be replaced by the word “indicate” or “represent” appropriately in its locations.
Response: Thank you for your careful check, the mistake in the paragraph has been corrected.
Section 4:
In paragraph 2, line 2, what is “CIC”? I think it means “Constant Intensity Cut”,but this abbreviation is not mentioned earlier. In line 3, the term “TeV to tens PeV, the in¬creasing trend as the shower energy is clear” should be rewritten as “TeV to tens of PeV, where the increasing trend of the attenuation length with the shower energy is clear”.
Response: Thank you for your careful check, the mistake in the paragraph has been corrected.
** Overall, the clarity of all figures should also be increased.
Response: All Figures have been increased.

---

## Editorial Decision

published